# ComSL: A Composite Speech-Language Model for End-to-End Speech-to-Text Translation

**Chenyang Le**[*]
Shanghai Jiao Tong University
Shanghai, China
`nethermanpro@sjtu.edu.cn`

**Yao Qian**
Microsoft Cloud and AI
Redmond, WA, US
`yaoqian@microsoft.com`

**Long Zhou**
Microsoft Research Asia
Beijing, China
`lozhou@microsoft.com`

**Shujie Liu**
Microsoft Research Asia
Beijing, China
`shujliu@microsoft.com`

**Yanmin Qian**
Shanghai Jiao Tong University
Shanghai, China
yanminqian@sjtu.edu.cn

**Michael Zeng**
Microsoft Cloud and AI
Redmond, WA, US
`nzeng@microsoft.com`

**Xuedong Huang**
Microsoft Cloud and AI
Redmond, WA, US
`xdh@microsoft.com`

## Abstract

Joint speech-language training is challenging due to the large demand for training data and GPU consumption, as well as the modality gap between speech and language. We present ComSL, a speech-language model built atop a composite architecture of public pretrained speech-only and language-only models and optimized data-efficiently for spoken language tasks. Particularly, we propose to incorporate cross-modality learning into transfer learning and conduct them simultaneously for downstream tasks in a multi-task learning manner. Our approach has demonstrated effectiveness in end-to-end speech-to-text translation tasks, achieving a new state-of-the-art average BLEU score of 31.5 on the multilingual speech to English text translation task for 21 languages, as measured on the public CoVoST2 evaluation set.[2]

## 1   Introduction

In recent years, an end-to-end (E2E) modeling approach has been widely applied to spoken language tasks, e.g., speech translation, speech summarization, speech understanding, etc. It enables us to train a single model by E2E optimization with spoken language task-oriented metrics. The conventional pipeline design generally contains two modules: speech model decodes the input speech into text and language model infers the recognized text to target language in translation task, topic sentence in summarization or intent in understanding task. These two modules are trained using their own respective criteria, which may not be optimal for each other. The cascaded process probably propagates errors that occurred in the current module to the following module. In addition, other information such as prosodic features contained in the speech signal, which are difficult to quantize using language symbols, can also be beneficial for spoken language tasks.

---

[*]The first author conducted this work during internship at Microsoft.
[2]The code is available at https://github.com/nethermanpro/ComSL.

37th Conference on Neural Information Processing Systems (NeurIPS 2023).

Unified speech-language pretraining based on Transformer architecture has largely boosted E2E modeling for spoken language tasks [46, 2, 31, 8, 7, 13, 11, 44]. The pretraining is conducted jointly on unlabeled speech, unlabeled text, paired speech-to-text and paired text-to-text in multiple languages by using both self-supervised and supervised learning. The unified representations from both speech and text modalities are simultaneously learned via shared model parameters or auxiliary modality matching losses in the pretraining stage. However, on the other hand, pretrained speech-only and language-only models are also becoming increasingly powerful. To mitigate modality gap and achieve the comparable performance of the cascaded module system, unified speech-language pretraining must leverage the same or larger scale of data used in speech-only or language-only model pertaining. This makes the training process very challenging.

In this paper, we propose **ComSL**: a **Com**posite **S**peech-**L**anguage Model for spoken language tasks. The main contributions of the paper can be summarized as follows:

1. Our composite model fully leverages existing pretrained models, eliminating the need for pre-training with large amounts of data from scratch. It can be directly fine-tuned for downstream tasks and demonstrates its data efficiency.

2. Our proposed cross-modality learning with speech-text mapping/matching on either representations or distributions is only based on the concatenation of paired speech and text. Unlike conventional approaches that use contrastive learning among the modalities, it does not require external or internal aligners to force-align speech and text at the token or word level. This simplifies the implementation and allows it to be incorporated into the fine-tuning stage.

3. We have conducted a comprehensive ablation study on bridging the gap of speech and language representations, including tasks, losses, and strategies, as well as comparisons with previous works.

4. Our model outperforms the SOTA Google USM model [44], Open AI Whisper model [26], and cascaded non-E2E models by 0.8, 1.8 and 1.0 average BLUE score improvements on the CoVoST 2 evaluation set, respectively.

## 2   Related Work

Early work for E2E Speech Translation (ST) mainly focused on the practicability of E2E approach as proof of the concept and the study of the network structure of Transformer for ST [9, 39, 28, 14, 15, 4]. But these models are unable to compete with cascaded models that are separately trained on abundant Automatic Speech Recognition (ASR) and Machine Translation (MT) data [29]. The recent achievements to tackle data scarcity in E2E ST can be broadly categorized into three directions: 1) pretraining [6, 30, 40, 42, 38, 1, 21]; 2) bridging modality gap [11, 23, 37, 33, 16, 43, 24, 46, 31, 45, 7, 13, 11, 44]; and 3) data augmentation [18, 19, 5, 20].

**Pretraining for E2E ST** Pretraining allows a model to be initially trained on either unlabeled data or labeled data from other tasks and thus enables the model be more effectively fine-tuned using labeled data from downstream tasks. The exploration of using pretrained model for E2E ST started with using either pretrained ASR model or MT model [6, 30, 40, 42]. However, neither of them is sufficient for downstream E2E ST. [38] proposed a curriculum pretraining method with an ASR loss as an elementary course, and a masked language model loss and a translation loss as two advanced courses. Most recently, speech-language model was jointly pretrained with a set of tasks including speech recognition, text translation, speech translation, and etc [46, 31, 8, 7, 13, 11, 44].

**Bridging modality gap for E2E ST** Text representations capture contextual semantics, syntax, and part-of-speech information, while speech representations capture contextual phonemes, speaker identity, prosody, and other acoustic characteristics. When jointly optimizing these two representations, there may be a gap between them due to their different focuses. Additionally, speech representations are typically longer than text representations because they are extracted at the frame level while text representations are extracted at the token level. To align text and speech representations or learn a mapping between them, text embeddings are up-sampled according to the corresponding phoneme duration in the paired speech. During training, a modality matching loss (MML), such as L2 loss, is applied between the up-sampled text representations and speech representations [11]. The mismatched length of speech and text representations was also addressed by repeating the text token input based on CTC results [37] and down-sampling speech representations to match the length of

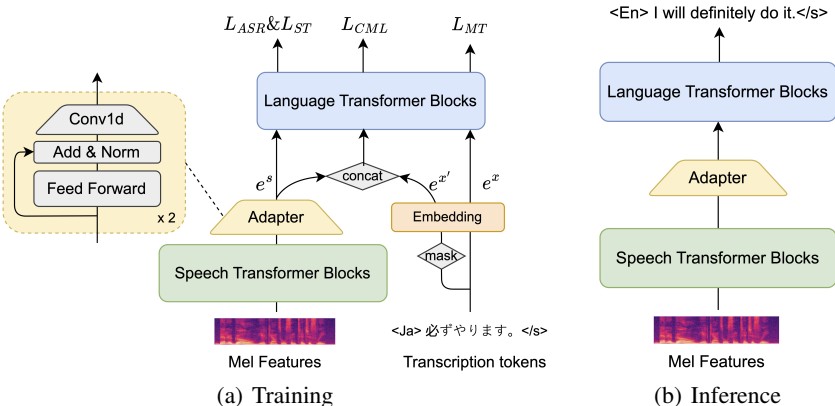

**(a) Training**          **(b) Inference**

Figure 1: Overall architecture of our model ComSL, consisting of speech Transformer Blocks, adapter, and language Transformer blocks. (a) model training with ASR, ST, MT and cross-modality learning (CML) tasks. (b) model inference only for ST tasks.

text representations [22]. Additionally, techniques such as mixing up speech and text features for input [16], leveraging hidden discrete units extracted from off-line speech-to-unit and text-to-unit generators [45], applying cross-attentive regularization to the representations from speech and text encoders [33], using contrastive learning at the word (WACO) and sentence (ConST) levels [24, 43], and employing knowledge distillation from MT to ST [23] have been shown to effectively bridge the gap between these two modalities.

**Data augmentation** ST training data, i.e., paired data of source language speech and target language text, is often limited. Data augmentation has been straightforwardly proposed to address this issue. Since the amount of paired speech recognition data is larger than that of ST, transcriptions of source language speech were converted into target language text using a high-performance MT system for ST training [41]. Additionally, source language texts from parallel machine translation corpora were synthesized into speech [19]. With forced alignment information, new ST paired data was generated by replacing suffix speech segments and translating combined transcription with MT. Furthermore, the ST training data was augmented by spec-augmentation [5] or mixing at frame, word and sentence levels [12].

## 3 Method

### 3.1 Problem formulation

End-to-end speech translation directly translates speech from a source language into text in a target language, without generating intermediate ASR transcription. It can be formulated to find the most likely label sequence $\mathbf{y} = \{y_1, y_2, ..., y_N\}$ (e.g., words or characters) of length $N$ in the target language given acoustic feature sequence $\mathbf{s} = \{s_1, s_2, ..., s_T\}$ (e.g., Mel Filter Bank features) of length $T$ in the source language. An ST training corpus $\mathcal{D}^{\mathrm{ST}} = \{(s, x, y)\}$ usually contains $\mathbf{x} = \{x_1, x_2, ..., x_M\}$, the transcription of speech in source language, in addition to $\mathbf{y}$ and $\mathbf{s}$. It allows us to use MT and ASR as auxiliary tasks in a multi-task learning manner during the optimization of E2E ST model, as

$$\mathcal{L} = \mathcal{L}_{ST} + \mathcal{L}_{MT} + \mathcal{L}_{ASR} \tag{1}$$

Multi-task learning (MTL) has been proven useful to share common knowledge among different tasks. However, it is non-rival in practice. ASR task can help improve the resolution of speech representations while the requirement of strict alignment between speech and text may negatively impact the performance of language pairs with big differences in word order. MT task can serve as a teacher to guide the ST task, as text-to-text mapping is relatively easier to learn than speech-to-text mapping. However, the mismatch between speech and text modalities may make this guidance less effective.

## 3.2 Model Architecture

Followed by previous work [32, 22, 33], our model is a composite model, which is mainly composed of three components: speech Transformer blocks, an adapter and language Transformer blocks stacked together, shown in Figure 1. All of the components except the adapter are initialized by pretrained models. We also leverage multi-task learning for both speech and text inputs in the training stage.

**Speech Transformer blocks** are initialized with the encoder parameters of Whisper model [26][3]. Whisper follows a standard sequence-to-sequence Transformer architecture [35] and has achieved strong performance on both speech-to-text recognition and translation tasks. It is trained on 680,000 hours of labeled speech recognition data as well as 125,000 hours of labeled X→en translation data. Here we only take its encoder as a powerful speech representation extractor.

**Adapter** is placed between the speech Transformer blocks and the language Transformer blocks. It contains two layers, in which each layer is composed of a feed-forward module and a 1-D convolution layer with stride two. It achieves four times down-sampling for speech encoder outputs in total. And the non-linearity of the feed-forward module can speed up the adaptation of speech representations to be fed into language Transformer blocks.

**Language Transformer blocks** are initialized with mBART model. mBART is composed of 12 layers of encoder and 12 layers of decoder with model dimension of 1024 on 16 heads. It is pretrained on monolingual data and fine-tuned on paired MT data and thus capable of many-to-many translation. All its parameters are used to enhance the translation capability of ComSL.

## 3.3 Cross-modality learning

In Equation 1, we can feed speech-only input and text-only input into the model to learn the tasks of ST, ASR, and MT, respectively. Additionally, we introduce cross-modality learning (CML) based on paired speech-text input to minimize the gap between speech and text modalities. Different from the previous speech-text alignment approaches that rely on externally forced-alignment methods to determine word or other unit boundaries in speech sequences, our approach intrinsically learns cross-modality information during model optimization. As illustrated in Figure 2, the speech embedding, $e^s$, from adapter and text embeddings, $e^x$, from text tokenizer are **concatenated** as input to mBART encoder, $\mathbf{Enc^t}$. After traversing the encoder, the concatenated form is **split** into $\hat{z^s}$ and $\hat{z^x}$ again to pass through mBART decoder, formulating

$$\hat{z^s}, \hat{z^x} = \mathbf{Enc^t}\left(e^s \oplus e^{x'}\right) \tag{2}$$

where $e^{x'}$ is the masked text embedding and $\oplus$ denotes concatenation. The following tasks are used for cross-modality learning.

**Masked Token Prediction (MTP)** The text tokens in the concatenated input are randomly masked by a probability $p_{mask}$. The MTP task is to predict the masked tokens in the output of the decoder. In practice we still let the decoder generate the whole sentence teacher-forcingly but we only add loss on the masked tokens.

$$\mathcal{L}_{MTP} = -\sum_{\mathcal{D}}\sum_{t=1}^{M}\sum_{v=1}^{|V|} \mathbf{1}(x_t = v \& x'_t = \text{<mask>}) * \log \mathcal{P}(x_t = v | x_{<t}, \hat{z^x}; \theta) \tag{3}$$

where $\mathcal{D}$ is the triplet corpus, $M$ is the transcription length, $V$ is the dictionary, $x$ is the transcription, $x'$ is the randomly masked transcription, $e_s$ is the speech embedding from the adapter, $\mathbf{1}(\cdot)$ is an indicator function. This task encourages the text encoder to fill in the missing token based on its speech corresponding through self-attention and helps to build a strong relation between two modalities.

**Speech to Text Mapping (STM)** This task is similar to a regular sequence-to-sequence mapping task (such as ASR or ST ) except that the encoder output hidden state is now conditioned on not only the input speech $s$ but also the masked ground-truth transcription $x'$. The STM task has two losses to

---

[3]https://github.com/openai/whisper. We use revision eff383b in our experiment.

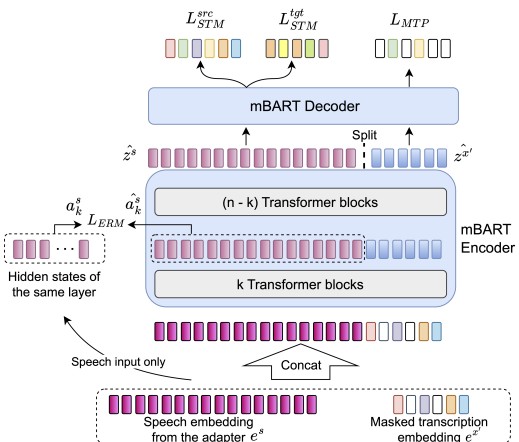

Figure 2: Illustration of cross-modality learning. The speech and text embeddings are concatenated into the mBART encoder for ERM loss and split into the decoder for STM and MTP tasks.

predict source transcription and target translation, respectively. Formally,

$$\mathcal{L}_{STM}^{src} = -\sum_{\mathcal{D}} \sum_{t=1}^{M} \sum_{v=1}^{|V|} \mathbf{1}(x_t = v) * \log \mathcal{P}(x_t = v | x_{<t}, \hat{z^s}; \theta) \tag{4}$$

$$\mathcal{L}_{STM}^{tgt} = -\sum_{\mathcal{D}} \sum_{t=1}^{N} \sum_{v=1}^{|V|} \mathbf{1}(y_t = v) * \log \mathcal{P}(y_t = v | y_{<t}, \hat{z^s}; \theta) \tag{5}$$

where $N$ is the translation length. By using this way, we can construct a hidden representation that is of the same length as the speech tokens but contains rich information learned via the attention on the ground-truth transcription. This hidden representation, $\hat{z}_s$, should have a better performance than the down-sampled representation, $e^s$, from the adapter on the top of the speech encoder. Our assumption is that the sequence of down-sampled representations has a similar length to that of text, making it more suitable as input for the text encoder. However, the underlying down-sampling operation may degrade its performance.

**Encoder Representation Matching (ERM)** To push the representation of speech-only input, $z^s$, closer to that of concatenate input, $\hat{z}_s$, we add a L2 loss between them. We do not directly add L2 loss between $z_s$ and $\hat{z}_s$ since the variation among them might be too large to learn well. Instead, we add the loss on the hidden state of the k-th Transformer block of the text encoder, $\hat{a_k^s}$ and $a_k^s$. k is a hyper-parameter and we set it to be 4 in our experiment.

$$\mathcal{L}_{ERM} = MSE\left(\hat{a_k^s}, a_k^s\right) \tag{6}$$

**Decoder Distribution Matching (DDM)** The performance of MT is generally better than that of ST especially when the input of MT is the ground-truth transcription, i.e., the human-labeled transcription of speech in the source language. We involve DDM into the loss of ST task as

$$\mathcal{L}_{ST} = -\sum_{\mathcal{D}} \sum_{t=1}^{N} \sum_{v=1}^{|V|} \left[ (1-\lambda_s) * \mathbf{1}(y_t = v) + \lambda_s * ||\mathcal{P}(y_t = v | y_{<t}, z^x; \theta)|| \right] * \log \mathcal{P}(y_t = v | y_{<t}, z^s; \theta) \tag{7}$$

where $\lambda_s$ is the weight for DDM and $|| \cdot ||$ means stop gradient. The modified ST loss allows ST task to learn not only from the ground-truth translation but also from the output distribution of MT task. The implementation of DDM is similar to the online knowledge distillation proposed in [33].

### 3.4 Training Strategies

#### 3.4.1 Fine-tuning Language Model

Before the model composition, we fine-tuned the language model with all the paired text data in the training data. The language model we integrated into our composite model is mBART-50 [34], which

is an extension of the original mBART model by adding 25 languages to support multilingual MT tasks in 50 languages. However, two languages, Welsh and Catalan, in our training corpus are still not covered by mBART-50. As we observed, mBART has a good extension capability so that it can achieve decent performance on these unseen languages by fine-tuning with limited data. We also tried to fine-tune the Whisper model before the model composition. Since Whisper has been trained with data in 100 languages, it has less impact on the final performance.

### 3.4.2 Multi-task Learning

We train ST, MT, ASR as well as CML by multi-task learning. For ASR task, we use a simple negative log likelihood loss:

$$\mathcal{L}_{ASR} = -\sum_{\mathcal{D}}\sum_{t=1}^{M}\sum_{v=1}^{|V|} \mathbf{1}(x_t = v) * \log \mathcal{P}(x_t|x_{<t}, z^s; \theta) \tag{8}$$

For CML, we average their losses from the decoder and add weight to the loss from the encoder:

$$\mathcal{L}_{CML} = 1/3 * (\mathcal{L}_{STM}^{src} + \mathcal{L}_{STM}^{tgt} + \mathcal{L}_{MTP}) + w_{ERM} * \mathcal{L}_{ERM} \tag{9}$$

Our final loss is the weighted sum of these tasks.

$$\mathcal{L}_{total} = w_{asr} * \mathcal{L}_{ASR} + w_{st} * \mathcal{L}_{ST} + w_{mt} * \mathcal{L}_{MT} + w_{CML} * \mathcal{L}_{CML} \tag{10}$$

### 3.4.3 Regularization on the MT Output

The language Transformer blocks (i.e., mBART-50) have been fine-tuned with MT tasks on our corpus before the model composition. To prevent MT task from overfitting in the multi-task learning for the composite model, we introduce an additional language model, i.e., fine-tuned mBART-50 model, $\theta'$, and freeze its parameters during training. We add a cross-entropy loss to minimize the difference of their output distributions. This operation is similar to that in ST task, i.e., using a better task as a teacher to guide a relatively worse task.

$$\mathcal{L}_{MT} = -\sum_{\mathcal{D}}\sum_{t=1}^{N}\sum_{v=1}^{|V|} \left[(1-\lambda_t)*\mathbf{1}(y_t = v) + \lambda_t*||\mathcal{Q}(y_t = v|y_{<t}, z^x; \theta')||\right] * \log \mathcal{P}(y_t = v|y_{<t}, z^x; \theta) \tag{11}$$

where $\mathcal{Q}$ is the output distribution of external fixed model and $\lambda_t$ is the weight for regularization.

### 3.4.4 Freezing Speech Encoder

The speech and language Transformer blocks are initialized by well-trained speech-only and text-only models, while the adapter is randomly initialized. These three components should use different learning rates during the training. To avoid it, we freeze the speech encoder at the first few epochs. This retains powerful speech representations during the early stage of fine-tuning when the gradients are unstable. The language component has the regularization on the MT output that plays a similar role.

## 4 Experiments

### 4.1 Datasets

**E2E ST Data** We conduct experiments on CoVoST 2 [36] dataset, a large-scale multilingual speech translation corpus covering translations from 21 languages into English and from English into 15 languages. It contains around 400 hours of English recordings and 900 hours of recordings from other 21 languages. Our work mainly focuses on X-EN, the non-English to English direction.

**Pseudo ST Data** The training data size for some low-resource language pairs in the CoVoST 2 dataset is limited. We add unlabeled translation data into the language directions that contain less than 30 hours of recordings into training in a self-training manner. Mozilla Common Voice [3] (version 11), a large-scale multi-lingual ASR dataset that is of the same source as CoVoST 2, is used to extract

data. We filter out all the data that is already in CoVoST 2 dataset to prevent repetition and test data leakage. Then we translate their transcriptions into English using the mBART model pre-finetuned on CoVoST 2 dataset. By this way, we add about 300 hours of recordings across 16 languages with paired speech and pseudo translation in text.

## 4.2 Experimental Setups

**Configuration** We build two versions of model, named **ComSL Medium** and **ComSL Large**. The difference among them is that the speech Transformer blocks are initialized with the encoder from Whisper in different sizes. The ComSL Medium uses Whisper medium encoder that contains 24 layers of Transformer blocks with 1024 hidden dimensions and 16 heads. The ComSL Large uses a Whisper large encoder that contains 32 layers of Transformer blocks with 1280 hidden dimensions and 20 heads. The two versions all have a two-layer convolution adapter and an mBART model that is initialized by a mbart50-large-mant-to-many-mmt checkpoint.[4] The total parameter size is about 0.9 billion for the medium version and 1.3 billion for the large version.

**Training** All the recordings are converted to 16 bit 16kHz mono-channel waveform for Mel Filter Bank features and then used as inputs to the model. During training, due to the memory issue, the input length of audio recording is limited to 11 seconds, which is long enough for cover 99 percent of our training data. Two techniques, deepspeed ZeRo [27] and activation checkpointing [10], are employed to optimize the usage of GPU memory. We save the checkpoint that has highest BLEU score on the validation set. It takes about 3 days to train on 4*8 Nvidia Tesla V100 GPUs with 32G of memory in each.

**Inference and Evaluation** During inference, we run beam search with beam size 5. For evaluation, we report detokenized corpus level BLEU scores on CoVoST 2 test set using sacreBLEU [25] .

## 4.3 Main Results

Our main results are shown in Table 1. We follow [7, 13] in dividing the 21 languages of the test set into three groups: High-resource (High), Medium-resource (Med), and Low-resource (Low). We report the average BLEU score for each group as well as the average for all 21 languages (Avg).[5] ComSL Medium and Large outperform Whisper Medium and Large, achieving BLEU scores of 29.7 and 30.1, respectively. The main factors contributing to this improvement are the replacement of the Whisper decoder with mBART and the application of our training strategies and losses in training the combined Whisper and mBART models. We observe that the performance of ComSL Medium is on par with that of Whisper Large, which has significantly more parameters. It is noteworthy that the BLEU scores for Whisper Medium and Large surpass those reported in the original paper [26], which shows zero-shot performance on the CoVoST testing set. The scores for Whisper models, listed in Table 1, result from fine-tuning Whisper with CoVoST 2 training data. Our BLEU score for the fine-tuned Whisper is higher than that shown in [44], which may be due to differences in the versions used for the experiments. When pseudo ST data is employed, both ComSL Medium and ComSL Large outperform the current SOTA, i.e., USM [44]. Our ComSL Large reaches a BLEU score of 31.5, which is 0.8 higher than USM with 0.7B fewer parameters. According to the data description in [44], the Common Voice corpus, where we extra data and generate pseudo translations, has also been included in USM training through self-supervised learning. Furthermore, our ComSL Medium and Large single models outperform Cascaded models that first conduct ASR with Whisper and then run MT with mBART-50 using ASR hypotheses as inputs. However, there is still a performance degradation due to ASR errors when compared to using ground-truth transcription as inputs for MT.

## 5 Analysis

### 5.1 Ablation study on training tasks/losses and strategies

We conduct an ablation study on training tasks/losses and strategies using ComSL Medium. The results, shown in Table 2, demonstrate how our strategies and tasks steadily improve overall perfor-

---

[4]https://huggingface.co/facebook/mbart-large-50-many-to-many-mmt

[5]The breakdown of BLEU scores in the different configurations on each of the 21 language pairs is listed in the Appendix.

| Model | PARAMS | High | Med | Low | Avg |
|---|---|---|---|---|---|
| mSLAM [7] | 2 B | 37.8 | 29.6 | 18.5 | 24.8 |
| Maestro [11] | 0.6 B | 38.2 | 31.3 | 18.4 | 25.2 |
| Mu2SLAM [13] | 0.6 B | 37.0 | 30.6 | 23.5 | 27.1 |
| Whisper Medium (Baseline Medium) | 0.8 B | 36.4 | 32.8 | 24.3 | 28.6 |
| Whisper Large (Baseline Large) | 1.6 B | 37.0 | 33.8 | 25.6 | 29.7 |
| USM [44] | 2 B | - | - | - | 30.7 |
| *Our model trained without pseudo ST data* | | | | | |
| **ComSL Medium** | 0.9 B | 37.4† | 34.1† | 25.3† | 29.7 |
| **ComSL Large** | 1.3 B | 37.7‡ | 34.5‡ | 25.8 | 30.1 |
| *Our model trained with pseudo ST data* | | | | | |
| **ComSL Medium** | 0.9 B | 37.3† | 35.6† | 26.6† | 30.8 |
| **ComSL Large** | 1.3 B | 37.6‡ | **35.9‡** | **27.5‡** | **31.5** |
| *Non-E2EST* | | | | | |
| Ground-truth Transcription + mBART-50 | 0.6 B | 42.1 | 39.9 | 35.8 | 38.0 |
| Whisper Medium + mBART-50 | 1.3 B | 37.3 | 34.5 | 26.4 | 30.4 |
| Whisper Large + mBART-50 | 2.2 B | 37.4 | 34.6 | 26.4 | 30.5 |

Table 1: Speech translation performance in terms of the average BLEU scores on non-English to English (xx-en) directions of the CoVoST 2 testing set. All models including Whisper and mBART-50 are fine-tuned with CoVoST 2 training set. The symbols "†" and "‡" are used to indicate that the model performance is significantly better (p<0.01) than the Baseline Medium and Large, respectively.

| Description | ST BLEU | ST Delta | MT BLEU |
|---|---|---|---|
| w/ ST task only | 26.48 | - | 29.13 |
| + MT task | 26.31 | -0.17 | 36.43 |
| + DDM | 27.87 | 1.56 | 36.21 |
| + ASR task | 28.29 | 0.42 | 35.92 |
| + Freeze encoder | 28.64 | 0.35 | 35.28 |
| + MT regularization | 29.40 | 0.76 | 37.40 |
| + CML Loss | 29.69 | 0.29 | 37.30 |
| + Pseudo data | 30.77 | 1.08 | 37.46 |

Table 2: Ablation study on the training strategies and tasks/losses. We start training the model with ST loss only and then adding a strategy or a task/loss over the previous experiment. We present the BLEU score of ST task, as well as the performance gain of ST task brought by each change.

mance. The model trained with only ST loss achieves a BLEU score of 26.48. Adding the MT task results in a slight drop to 26.31, indicating that directly applying the MT task has little impact on performance. However, when DDM is leveraged, the BLEU score increases significantly to 27.87. The ASR task and corresponding training strategy of freezing speech encoder parameters in the early stage of training improve the BLEU score by 0.42 and 0.35, respectively. MT regularization, which prevents overfitting in the MT task, enhances performance by an additional 0.76 BLEU score. CML, which matches speech and text representations, boosts performance to BLEU score of 29.69. Finally, using pseudo ST data further increases the BLEU score to 30.77. To further investigate the performance of the integrated language model in ComSL, we feed ground-truth speech transcription to it, even though ComSL only requires speech as input for the ST task. The corresponding BLEU score of the MT task shown in Table 2 indicates that our training strategies and tasks continue to reduce the degradation of MT performance from mBART-50, finally achieving a BLEU score of 37.46 compared to 38.0 for Ground-truth Transcription + mBART-50 shown in Table 1.

### 5.2 Comparison among different cross-modality learning (CML) methods

We compare our cross-modality learning methods with previous works. The model trained with all strategies and tasks except the methods of minimizing modality gap refers to the baseline. The performance of the baseline and those with our CML method and previous MML, ConST, and WACO methods are listed in Table 3. Our CML reaches 29.69 BLEU score, the best one among all listed methods in the table. MML also performs well, achieving a BLEU score of 29.6, slightly lower than ours, but it requires internal alignment at the token level. Calculation of alignment by

| Method | Forced-Alignment | Avg BLEU |
|--------|------------------|----------|
| Baseline | N | 29.40 |
| MML [11] | Token level | 29.60 |
| ConST [43] | N | 29.18 |
| WACO [24] | Word level | 29.37 |
| Our CML | N | 29.69 |

Table 3: Comparison between different methods for minimizing modality gap on ComSL Medium. The second column shows what kind of forced-alignment is needed for the method. N denotes 'Not Needed'.

whatever internal or external token-level forced aligner, adds a burden to the training process. In our implementation, contrastive-based methods (ConST and WACO) fail to surpass the baseline model. We conjecture that it may be due to the nature of contrastive loss, which pulls negative and positive samples apart, which may cause an undesirable gradient direction for a supervised pretrained model like Whipser.

We also take an in-depth look at the hidden level to see what happens after applying cross-modality learning tasks. Figure 3 shows the similarity matrices of speech and text representations. These representations are taken from the output of the 4th layer in the mBART encoder. A higher number, indicated by a darker color in the figure, means that the representations are more similar to each other. Here are our findings, 1) Our model can match text and speech representations fairly well even without explicitly applying any modality gap minimizing methods, as shown in Figure 3-a). We believe this is most likely due to the use of knowledge from ASR task, which leaves less room for the CML method to further improve performance. 2) The model trained with our CML can produce a unified representation not only for speech and text but also for non-speech tokens like silence. As illustrated by the green box in the figure, non-speech tokens like punctuation (period) and silence are aligned much better than those without CML. This is a unique phenomenon observed in our approach. (We tried plotting matrices using other methods listed in Table 3, but all looked like the left figure.) 3) Our CML can help encode speech information more precisely. For example, in the blue box area, there is a compound two-syllable word with a very short pause in between. Our model better identifies the short pause and sharpens the distributions of surrounding tokens.

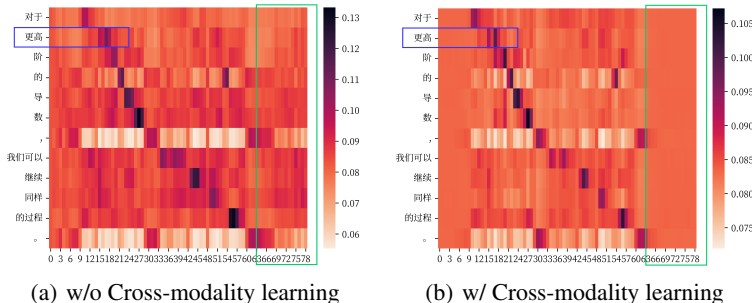

(a) w/o Cross-modality learning      (b) w/ Cross-modality learning

Figure 3: Similarity matrices of speech hidden states and text hidden states(softmax alone the text axis). These hidden states are taken from the output of the 4-th text encoder layer.

## 6    Conclusions, Limitations and Future Work

In this paper, we presented a Composite Speech and Language (ComSL) model to perform E2E speech-to-text translation tasks. We bridged the gap between speech and text representations through cross-modality learning tasks, in addition to other auxiliary tasks like speech recognition and machine translation. Our composite model outperformed the constituent speech model (Whisper) or cascaded speech and language models (Whisper+mBART) in a pipeline manner. By adding pseudo ST data, our model achieved new SOTA performance on the public CoVoST2 evaluation set. However, our approaches have not been exploited on decoder-only based Large Language Models (LLMs) in this study due to their recent public availability and the requirement of large GPU resources. Additionally,

our model has only been evaluated on speech-to-text translation tasks due to the limitations of currently available E2E datasets. In the future, we plan to combine our approaches with Low-Rank Adaptation (LoRA) [17] or other light-weight adapters and use efficient data augmentation techniques to fine-tune a large composite model for various downstream spoken language tasks.

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
