# OpenReview forum: "ComSL: A Composite Speech-Language Model for End-to-End Speech-to-Text Translation"
_NeurIPS.cc/2023/Conference — NeurIPS 2023 poster_

### Official Review · Reviewer_nRjc · 2023-06-28

**Soundness:** 3 good
**Presentation:** 3 good
**Contribution:** 2 fair
**Rating:** 6
**Confidence:** 5

**Summary:**

Authors propose a speech-text pretrained model for spoken language tasks which leverages already existing pre-trained speech and language models. Modality mapping/alignment is based on a concatenation of paired speech-text (no need of word level alignment). The model is evaluated on a speech-to-text translation (S2TT) task (on CoVoST2 dataset) and slightly outperform previous models such as Whisper for S2TT.






**Strengths:**

-Good performance reported on CoVoST speech-to-text translation tasks

-An approach that leverage existing pre-trained models (whisper speech encoder; mBART text2text) to build efficient ST systems



**Weaknesses:**

-Difference/positioning related to previous speech-text pretrained models should be improved: in what aspect does your method really differ from SpeechT5, SLAM, etc ?

-Improvement over Whisper Large when model is trained without pseudo ST data is tiny (and maybe not significant) so does the improvement really comes from the new approach/losses proposed or does it come only from the use of pseudo ST data ?

-Only Whisper speechencoder is used, why ? Authors could have also chosen wav2vec multilingual (XLS-R) or HuBERT speech encoders

**Questions:**

-In what aspect does your method really differ from SpeechT5, SLAM, etc ?

-"Speech Transformer blocks are initialized with the encoder parameters of Whisper model" why using Whisper speech encoder only and not XLS-R (multilingual wav2vec) or HuBERT ?

-"Different from the previous speech-text alignment approaches that rely on externally forced-alignment methods to determine word or other unit boundaries in speech sequences," => Do SpeechT5 & SLAM need external word-level forced-alignment (i don't think so)

-(tab 1) does the improvement really comes from the new approach/losses proposed or does it come only from the use of pseudo ST data ?

**Limitations:**

Limitation mentioned are the use of mBART encoder decoder instead of large decoder only models (with bigger architectures)

---

> ### Author Rebuttal · Authors · 2023-08-08
>
> Thank you for your insightful questions. We have provided our responses right after each question.
>
> 1.	In what aspect does your method really differ from SpeechT5, SLAM, etc ?
>
> Answers:
> Our methods differ from those used in SpeechT5, SLAM, etc. in the following ways:
>
> A) The cross-modality learning (CML) is employed in different training stages. In previous works such as SpeechT5 and SLAM, CML is implemented in the pre-training stage, while ours is used in the fine-tuning stage together with fine-tuning task losses.
>
> B) Different CML methods/objectives are utilized. In SpeechT5, its technique for joint pre-training involves utilizing vector-quantized embeddings as a bridge to align the speech and text representations through a shared codebook and mixing up contextual representations and quantized latent representations to guide the quantizer to use cross-modal information. In SLAM, although it also includes a joint input of speech and text, it predicts masked text or speech spans with BERT or w2v-BERT on top of the encoder.
>
> In contrast, our approach does not require quantization for speech input or an MLM loss for encoders. Therefore, our approach is completely different from SpeechT5 and SLAM. Our approach is more flexible, requires less data as it is applied during the fine-tuning stage, and can be used for adaptation between any composite transformer-based speech and language model.
>
> 2.	"Speech Transformer blocks are initialized with the encoder parameters of Whisper model" why using Whisper speech encoder only and not XLS-R (multilingual wav2vec) or HuBERT ?
>
> Answers: We believe that our methods are also applicable to XLS-R or HuBERT. HuBERT was pre-trained using Libri-light (60k hours), a collection of spoken English audio without labels. As a result, HuBERT is not a multi-lingual model and requires much more multi-lingual fine-tuning data to produce decent translation performance. On the other hand, XLS-R was pre-trained on massive multi-lingual data in a self-supervised learning manner. However, Whisper prevails in multi-lingual tasks recently since it was trained with much more supervised data and achieved better performance. We chose to fine-tune Whisper as a very strong baseline and explore the potential of our approaches. Considering that Whisper uses a standard transformer architecture, we believe it is representative of transformer-based speech encoders. If given adequate computational resources, we will try HuBERT or XLS-R as well in the future.
>
> 3.	"Different from the previous speech-text alignment approaches that rely on externally forced-alignment methods to determine word or other unit boundaries in speech sequences," => Do SpeechT5 & SLAM need external word-level forced-alignment (i don't think so)
>
> Answers:  You are correct. SpeechT5 and SLAM do not require external word-level forced alignment. We will rephrase this sentence to make it clear that it refers to the MML loss in USM and the word-aligned contrastive loss in WACO, which are used for comparisons in Table 3.
>
> 4.	(tab 1) does the improvement really comes from the new approach/losses proposed or does it come only from the use of pseudo ST data?
>
> Answers: The improvement comes from both the new approach/losses and the use of pseudo ST data. Please refer to the attached PDF file in the “global” response to all reviewers jointly for the results of significance tests, which show that, regardless of whether pseudo ST data is added, our ComSL Medium/Large models significantly outperform their corresponding baseline Medium/Large models (i.e., finetuned Whisper) on high-resource and mid-resource languages, with a significance level of p<0.01. In addition, we did not use any pseudo ST data for high-resource languages. For more information on the statistics of the pseudo data, please refer to the appendix.

---

> > ### Comment · Reviewer_nRjc · 2023-08-16
> > **updated score after rebuttal**
> >
> > hi
> >
> > tks for your detailed answers to my questions/comments
> > i updated my score from 5 to 6 (Weak Accept)
> >
> > best

---

> > > ### Author Response · Authors · 2023-08-16
> > >
> > > We are delighted that our response addressed your concerns and are grateful for your insightful feedback and the updated score.

---

### Official Review · Reviewer_7Di5 · 2023-07-06

**Soundness:** 3 good
**Presentation:** 3 good
**Contribution:** 3 good
**Rating:** 6
**Confidence:** 4

**Summary:**

This paper proposes a composite speech-language model for speech-to-text (ComSL) translation. ComSL first leverages existing pre-trained models for initialization, including Whisper speech recognition model and mBART machine translation model. And then proposed several modality alignment methods that do not require additional tools. Extensive experiments on CoVoST2 demonstrate the superiority of the proposed method, which also achieves a new SOTA performance.

**Strengths:**

1. This paper provides insight into designing large-scale spoken language models.
2. This paper proposes several methods based on concatenated embeddings to align the representation of speech and text.
3. The proposed method outperforms previous work and achieves a new SOTA performance.

**Weaknesses:**

1. The training process may be unstable. As shown in the appendix, multiple losses are distributed by various weights from 0.1 to 0.8. The authors do not explain how these hyperparameters are determined, and how much the impact is. A solid system is expected.
2. Although this paper achieves a new SOTA performance, it mainly relies on the combination of several common techniques. Therefore, the contribution is limited. To achieve SOAT performance, ComSL is trained with pseudo ST data, resulting in unfair comparisons.

**Questions:**

See weaknesses

---

> ### Author Rebuttal · Authors · 2023-08-08
>
> Thank you for your insightful questions. We have provided our responses right after each question.
>
> 1.	The training process may be unstable. As shown in the appendix, multiple losses are distributed by various weights from 0.1 to 0.8. The authors do not explain how these hyperparameters are determined, and how much the impact is. A solid system is expected.
>
> Answers:  Thank you for bringing this to our attention. We will make sure to clarify it in the revised version of our paper. Unlike training from scratch, our ComSL model weights were initialized with pre-trained speech and language models, which greatly stabilized the training process. Although we employed various weights for multiple losses, we set most weights empirically or referred to values used in other studies. Our consideration for relatively high or low weights depended on the contributions of the losses. For example, we set a high weight for DDM since the performance gap between AST and MT was large, and relatively smaller weights for other tasks like MT since MBart had already been fine-tuned before being combined with Whisper. We also tried tuning the weights within a small range, such as DDM from 0.8 to 1, and observed that its impact on performance was marginal. We did not perform a “brute force” hyperparameter tuning as it was unaffordable.
>
> 2.	Although this paper achieves a new SOTA performance, it mainly relies on the combination of several common techniques. Therefore, the contribution is limited. To achieve SOAT performance, ComSL is trained with pseudo ST data, resulting in unfair comparisons.
>
> Answers:
>
> a)	Our contributions are outlined in the final paragraph of the introduction section. With the emergence of pre-trained speech and language models, we proposed an approach that combines two publicly available models and fine-tunes them directly for downstream tasks, along with cross-modality learning, which is typically used during the pre-training stage. We hope that our approach will offer a new avenue for research in academia, where large model pre-training is not feasible due to limited computational resources.
>
> b)	The improvement of ComSL comes from both the new approach/losses and the use of pseudo ST data. Please refer to the attached PDF file in the “global” response to all reviewers jointly for the results of significance tests, which show that, regardless of whether pseudo ST data is added, our ComSL Medium/Large models significantly outperform their corresponding baseline Medium/Large models (i.e., finetuned Whisper) on high-resource and mid-resource languages, with a significance level of p<0.01. Nowadays, it is very challenging to conduct apple-to-apple comparisons due to differences in the amount of training data and model architecture/size. Most leaderboards/benchmarks do not require the listing of training data. Additionally, the Pseudo ST data we used was extracted from the Common Voice corpus, which was also included in USM pre-training in the form of self-supervised learning, according to the data description in the USM paper.

---

> > ### Author Response · Authors · 2023-08-20
> >
> > Dear reviewer 7Di5,
> >
> > We greatly appreciate your time reviewing our paper.  We hope that our rebuttal has addressed your concerns.  If you have any further questions or comments, we would be more than happy to provide clarification.
> >
> > Thanks!
> >
> > Authors

---

> > ### Comment · Reviewer_7Di5 · 2023-08-21
> >
> > I appreciate the detailed explanations. I will update the score from 5 to 6.

---

### Official Review · Reviewer_KhJu · 2023-07-07

**Soundness:** 3 good
**Presentation:** 3 good
**Contribution:** 2 fair
**Rating:** 6
**Confidence:** 5

**Summary:**

This work presents a speech-language model built from both speech-only and language-only pretrained models. By compositing pre-trained models from 2 modalities, the authors show that a data-efficiency for spoken language tasks can be achieved. In particular, the authors proposed a few cross-modality loss functions which are designed to  build a strong relationship between 2 modalities. The authors demonstrated that their method is able to achieve a new sota BLEU on the CoVoST2 evaluation task.

**Strengths:**

This paper is well written. It explains the motivations and methods clearly. Besides great overall results, it also presents a detailed analysis including ablation study and in-detail examination of usage of the cross-modality.

**Weaknesses:**

- It is a bit disappointing to see from the result the cross modality tasks, which is the main contribution in this work, does not give significant difference in terms of BLEU score (Table 2, 29.40 --> 29.69 due to CML loss). I believe the follow up improvement from pseudo data can be obtained without CML loss.

- It would be better if the authors can clearly state how many parameters are in the pre-trained ASR encoder, the LM (mBART) and in the adapters; and which parts are finetuned or fixed during training.

- Though the model is trained with both ASR and AST loss, the authors only evaluate  the model on the AST / MT tasks. It would be great if the author can provide a benchmark on ASR as well.

- It is not immediately clear to me that whether the comparison of this work to USM / whisper is fair. It seems to be both whisper and USM does not finetune on CoVoST2 tasks, while they both use CoVoST2 tasks as an out-of-domain evaluation. On the other hand, whisper is able to perform both ASR and AST using the same model with just different prompt, while the authors did not reveal their ASR performance.

There are also a recent paper relevant to this work:

Rubenstein, Paul K., et al. "AudioPaLM: A Large Language Model That Can Speak and Listen." arXiv preprint arXiv:2306.12925 (2023).

Understandably, this paper appears after NeuralPS submission deadline.  The authors may consider to cite it since it achieves a new SOTA on CoVoST2.

**Questions:**

- After the finetuning on CoVoST2, I think the model can still perform ASR ? But it would be great that the authors to clearly state that (or explain why it cannot).

**Limitations:**

I think the authors have addressed limitations adequately.

---

> ### Author Rebuttal · Authors · 2023-08-08
>
> Thank you for your insightful questions. We have provided our responses right after each question.
>
> 1.	After the finetuning on CoVoST2, I think the model can still perform ASR ? But it would be great that the authors to clearly state that (or explain why it cannot).
>
> Answers: You are correct. The ComSL can perform ASR after the finetuning on CoVoST2. Due to space limitations in the full paper, we reported the ASR performance in the appendix enclosed in the supplementary materials, along with the ST and MT tasks for each of the 21 languages in the CoVoST 2 evaluation set. We also included statistics on pseudo data and experimental details in the appendix. If the lack of an evaluation of the ASR task affected your score on our paper, we hope you will consider adjusting it.
>
> Table 6 in the appendix shows that ComSL outperforms finetuned Whisper on the ASR task for some languages, but its overall performance is slightly worse. This phenomenon has also been observed in other studies, and as a result, pretrained models are often finetuned for the ASR task only (instead of multi-task such as both ASR and AST tasks) to achieve better performance, as was done on USM (Reference [44]).
>
> 2.	It is a bit disappointing to see from the result the cross modality tasks, which is the main contribution in this work, does not give significant difference in terms of BLEU score (Table 2, 29.40 --> 29.69 due to CML loss). I believe the follow up improvement from pseudo data can be obtained without CML loss.
>
> Answers:
> We have done a significance test on comparing with/without CML, which shows CML has a significant impact on performance for high-resource and mid-resource languages with a significance level of p<0.05, but not on low-resource languages. During training, multiple tasks, including AST, ASR, MT, and CML, are conducted. The shared model parameters are updated not only using the speech task but also the text task. This reduces the modality gap between speech and text to a certain extent, and as a result, the CML loss might not be as significant as expected, especially after the DDM (AST and MT distribution matching) loss was employed as complementary method for reducing the modality gap. In addition, we suspected that limited training data of low-resource languages might be causing issues, such as limited vocabulary coverage, which could affect the efficiency of our proposed methods.
> Our experimental results show that the CML loss can significantly improve performance on high-resource languages, while adding more pseudo-ST data after applying the CML loss does not provide additional benefits. However, we have not yet tested the alternative approach of adding pseudo-ST data first and then applying the CML loss on low-resource languages. We will consider conducting this comparison in future research.
>
> 3.	It would be better if the authors can clearly state how many parameters are in the pre-trained ASR encoder, the LM (mBART) and in the adapters; and which parts are finetuned or fixed during training.
>
> Answers: The parameter sizes for different components are as follows: 1)Whisper medium encoder: 0.3B; 2) Whisper large encoder: 0.7B; 3)mBART (encoder+decoder): 0.6B; 4) Adapter: 30M. During the first third of the training steps, we only fixed the parameters of pre-trained ASR encoder (i.e., Whisper encoder). Other than that, all parameters were fine-tuned.
>
> 4.	Though the model is trained with both ASR and AST loss, the authors only evaluate the model on the AST / MT tasks. It would be great if the author can provide a benchmark on ASR as well.
>
> Answer is the same as that of first question.
>
> 5.	It is not immediately clear to me that whether the comparison of this work to USM / whisper is fair. It seems to be both whisper and USM does not finetune on CoVoST2 tasks, while they both use CoVoST2 tasks as an out-of-domain evaluation. On the other hand, whisper is able to perform both ASR and AST using the same model with just different prompt, while the authors did not reveal their ASR performance.
>
> Answers:  Actually, the BLEU scores for Whisper and USM reported in Table 1 were achieved by finetuning the models on the CoVoST 2 training set. We finetuned Whisper models and achieved BLEU scores of 28.6 and 29.7 for Whisper medium and Whisper large, respectively, which are higher than those reported in the original Whisper paper (Reference [26]). As for the USM model, since it is not a public model, we cited the BLEU scores from its paper (Reference [44]). Section 3.3.2 of this paper indicates that the USM model evaluated on CoVoST2 was finetuned using the CoVoST2 training set and text translation data such as WMT or TED talks as available. Therefore, our comparison is fair in the context of in-domain finetuning/evaluation. Additionally, we reported the ASR performance in the appendix enclosed in the supplementary materials.
>
> 6.	There are also a recent paper relevant to this work: Rubenstein, Paul K., et al. "AudioPaLM: A Large Language Model That Can Speak and Listen." arXiv preprint arXiv:2306.12925 (2023). Understandably, this paper appears after NeuralPS submission deadline. The authors may consider to cite it since it achieves a new SOTA on CoVoST2.
>
> Answers: Thank you for bringing this to our attention. We will be sure to cite this most recent and relevant work on leveraging LLMs to further improve performance in our paper.

---

> > ### Comment · Reviewer_KhJu · 2023-08-20
> >
> > Thanks very much for the detailed response and explanation. I am satisfied with the author's response.

---

> > > ### Author Response · Authors · 2023-08-20
> > >
> > > We are joyed that our response has successfully addressed your concerns and appreciative of your valuable feedback.

---

### Official Review · Reviewer_tyJC · 2023-07-08

**Soundness:** 3 good
**Presentation:** 2 fair
**Contribution:** 3 good
**Rating:** 6
**Confidence:** 4

**Summary:**

With the goal to improve speech translation task, this work, leverages a multi-task training approach optimizing weighted sum of ASR, ST, MT and cross-modality learning (CML) objectives. Using mBART encoder and decoder Transformer blocks, the CML objective is purposed to better align speech and text modalities. CML involves a masked token prediction (MTP), speech to text mapping (STM), and encoder representation matching (ERM) objectives.

Overall, the proposed arc (ComSL) includes a speech and textual transformer blocks, and a two layered bride adapter module. Before the multi-task step, pre-trained models (mBART and Whisper) are fine-tuned using training data for 50 languages, mainly involving CoVoST2 data set. Results are provided for low, medium and high-resource languages show ComSL's performance improvements over recently proposed multi-modal (text and speech) models such as USM and Whisper.

**Strengths:**

- This work, is well motivated suggesting an cross-modality learning objectives, to address the modality gap in the next frontier for a working composite speech and textual model.
- The results for ST are encouraging, performing similarly or slightly better than recently proposed multi-task based speech-text models like Whisper and USM.

**Weaknesses:**

- Although, ComSL is well motivated to focus on bridging the modality gap, the pre-training and fine-tuning using in-domain data doesn't seem to deliver significant improvements. With close results, and lack of significance testing, its rather difficult to interpret if the proposed approach is better performing.
- Given the focus on multi-task learning, the provided evaluation (only for ST task) is unexpected. Adding at least ASR task in to the comparison could have provided more substance to the report.
- Discussion and analysis part could have focused on ST cases (or examples) where CML improves over other models without using the cross-modality objectives. This would show the importance of introducing the objective.

**Questions:**

- What was the motivation for not evaluating other tasks, as this model is trained in a multi-task setting, at least the ASR evaluation makes sense.
- Given the number of objectives involved, particularly for the CML part, how does model training and inference complexity compares?
- Table 1, all models ... are fine-tuned with CoVoST2 data, does this include USM model? if not, what is the justification for a fair comparison?
- Table 1, for Non-E2EST, does it make more sense to use Whisper + and a standard MT model trained separately?
- Given most numbers are quite close, in table 1-3, adding statistical significance could make the results more meaningful, and to the support SOTA assertion.
- L103 fix unfinished sentence
- S3.3 how does the split is formulated for z^s and z^x before passing it to mBART decoder? Obviously the split is necessary to optimize the MTP objective using z^x, have you considered directly using e^x without the concat step?

**Limitations:**

Limitations are included, reflecting the main content of the work. I suggest authors to consider a separate limitation part and to consider societal impact if available.

---

> ### Author Rebuttal · Authors · 2023-08-08
>
> Thank you for your insightful questions. We have provided our responses right after each question.
>
> 1.	What was the motivation for not evaluating other tasks, as this model is trained in a multi-task setting, at least the ASR evaluation makes sense.
>
> Answers:
> Due to space limitations in the full paper, we have reported the ASR performance in the appendix of the supplementary materials. If the lack of evaluation on the ASR task affected your score, we hope you will consider adjusting it.
> Table 6 in the appendix shows that ComSL outperforms finetuned Whisper on the ASR task for some languages, but its overall performance is slightly worse. This phenomenon has also been observed in other studies, and as a result, pretrained models are often finetuned for the ASR task only (instead of multi-task such as both ASR and ST tasks) to achieve better performance, as was done on USM (Reference [44]).
>
> 2.	Given the number of objectives involved, particularly for the CML part, how does model training and inference complexity compares?
>
> Answers:
> During training, objectives other than speech-to-text translation loss are included. However, these additional objectives are not necessary during the inference stage, so the complexity remains the same as a conventional speech-to-text translation model. Our observations show that training time increases by a factor of 1.2, from 2.5 hours per epoch to 3 hours per epoch using 32 V100 GPUs, when compared to a model trained without CML losses.
>
> 3.	Table 1, all models ... are fine-tuned with CoVoST2 data, does this include USM model? if not, what is the justification for a fair comparison?
>
> Answers:
>  Yes, the USM model has been finetuned. USM is not a public model, but we have cited the BLEU scores from its paper (Reference [44]). Section 3.3.2 of this paper indicates that the USM model evaluated on CoVoST2 was finetuned using the CoVoST2 training set and text translation data as available. Additionally, USM for ASR and ST tasks were finetuned separately, so the results of the ASR and ST tasks shown in the USM paper were obtained from different finetuned USM models.
>
> 4.	Table 1, for Non-E2EST, does it make more sense to use Whisper + and a standard MT model trained separately?
>
> Answers:
> We are not entirely sure if we understand your question correctly. The mBART-50 model (please check footnote 3 on page 7) that we used is already a MT model. We finetuned this model on CoVoST2 text data. For the first row under Non-E2EST in Table 1, we fed the ground truth transcription to the mBART model to measure its performance, simulating a cascade system with a perfect ASR model output. For the second and third rows, we used Whisper as the ASR model for fair comparison. To the best of our knowledge, mBART-50 is the best pretrained translation model on CoVoST text data, even better than most LLMs (most LLMs are not finetuned for translation).
>
> 5.	Given most numbers are quite close, in table 1-3, adding statistical significance could make the results more meaningful, and to the support SOTA assertion.
>
> Answers:
> Thank you for your suggestions. We have done significance tests for the results in table 1-3.
>
> For Table 1, please refer to the attached PDF file in the “global” response to all reviewers. Our tests show that, regardless of whether pseudo ST data is added, our ComSL Medium/Large models significantly outperform their corresponding baseline Medium/Large models (i.e., finetuned Whisper) on high-resource and mid-resource languages, with a significance level of p<0.01. For low-resource languages, where the training data for most languages is around or less than two hours, the addition of pseudo ST data has a significant impact on improving ST performance. We suspected that limited training data for a specific language might be causing issues, such as limited vocabulary coverage, which could affect the efficiency of our proposed methods. This is why we added pseudo ST data only to mid-resource and low-resource languages. Please refer to the appendix for statistics on the pseudo data.
>
> Table 2 shows that adding only the MT task can negatively impact performance. So we conducted significance tests for all tasks/losses except for the MT task. The test results show that the CML loss can significantly improve ST performance on high-resource and mid-resource languages, with a significance level of p<0.05, but not on low-resource languages. All other tasks/losses significantly improve performance on all languages.
>
> For Table 3, our significance tests show that minimizing the modality gap has a significant impact on performance for high-resource and mid-resource languages. However, the differences between various methods of modality gap minimization are not significant. In our multi-task learning approach, we conducted several tasks, including AST, ASR, MT, CML, and others. The shared model parameters are updated using both speech and text tasks, which reduces the modality gap between speech and text to a certain extent. As a result, the explicit modality gap minimization loss may not be as significant as expected, especially after the DDM (AST and MT distribution matching) loss was employed as a complementary method for reducing the modality gap.
>
> 6.	L103 fix unfinished sentence
>
> Answer:
> We will fix it.
>
> 7.	S3.3 how does the split is formulated for z^s and z^x before passing it to mBART decoder? Obviously the split is necessary to optimize the MTP objective using z^x, have you considered directly using e^x without the concat step?
>
> Answers:
> Yes, we have considered this. The mBART used in our ComSL has been finetuned using text from the CoVoST 2 training set, so the improvement from adding a text task in ComSL training is marginal. For the MTP task, the goal is to match speech and text representations through self-attention. If we were to directly use e^x, it would become a pure text denoising task and would reduce the contribution to minimize modality gap.

---

> > ### Author Response · Authors · 2023-08-20
> >
> > Dear Reviewer tyJC,
> >
> > We appreciate the time and effort you have put into reviewing our paper.  It is our hope that our rebuttal has successfully addressed all of your concerns. We would like to draw your attention to the fact that we had provided the ASR evaluation results in the appendix of the supplementary materials. We sincerely hope that this clarification will positively influence your opinion on the quality of our submission. If you have any further questions or comments, we would be more than happy to provide further clarification.
> >
> > Thank you!
> >
> > Authors

---

### Author Rebuttal · Authors · 2023-08-08

We would like to thank all reviewers for taking time and effort to review our paper. We appreciate all your valuable comments and suggestions. To the common concerns and questions, our responses are listed here.

1. ASR evaluation

Due to space constraints in the full paper, we have included the ASR performance results in the appendix of the supplementary materials. This appendix also contains the ST and MT task results for each of the 21 languages in the CoVoST 2 evaluation set. Please review the ASR results in the supplementary materials. We will consider including a summary of these results in the revised version of the full paper. If the absence of an evaluation of the ASR task had an impact on the score you gave our paper, we hope you consider adjusting it accordingly.

Calculating WER for multi-lingual ASR outputs depends on the methods of text normalization and word segmentation for some languages. We were unable to find a standard and publicly available tool for this purpose, so we developed our own method and used it to compare the ASR performance of the models we built, namely Whisper Large finetuned and ComSL Medium/Large. It is important to note that the WER numbers we reported cannot be directly compared with those reported in other publications.

2. Statistical significance test

We have updated Table 1 to include the results of our statistical significance tests, which can be found in the attached PDF file. Our tests show that, regardless of whether pseudo ST data is added, our ComSL Medium/Large models significantly outperform their corresponding baseline Medium/Large models (i.e., finetuned Whisper) on high-resource and mid-resource languages, with a significance level of p<0.01. For low-resource languages, where the training data for most languages is around two hours, the addition of pseudo ST data has a significant impact on improving ST performance. We suspected that limited training data for a specific language might be causing issues, such as limited vocabulary coverage, which could affect the efficiency of our proposed methods. This is why we added pseudo ST data only to mid-resource and low-resource languages. Please refer to the appendix for statistics on the pseudo data.

3. Main contributions/differences

In addition to the main contributions summarized in the last paragraph of the introduction section, we would like to highlight our proposed approach that combines two publicly available pre-trained speech and language models and fine-tunes them directly for downstream tasks. This approach also incorporates cross-modality learning, which is typically used during the pre-training stage. We believe that our approach offers a new avenue for research in academia, where large model pre-training may not be feasible due to limited computational resources.

---

> ### Author Response · Authors · 2023-08-16
>
> We would like to express our gratitude to the reviewers for taking the time to review our manuscript again. As we approach the end of the authors-reviewers discussion period, we welcome any remaining concerns and are happy to address them at the reviewers’ request.

---

### Decision · Program_Chairs · 2023-09-21

**Decision:**

Accept (poster)

**Comment:**

This research introduces ComSL, a composite speech-language model, which integrates pre-trained speech and text models to enhance speech translation. By incorporating cross-modality learning objectives and alignment methods without additional tools, the model achieves improved data efficiency and forms a stronger inter-modality relationship. Extensive evaluations on the CoVoST2 dataset indicate that ComSL surpasses existing benchmarks, setting a new state-of-the-art performance.

Many concerns were addressed in the rebuttal. The main critical review was from tyJC, who praised the method but had concerns about the evaluation, namely the lack of ASR results. The authors provided ASR results in the rebuttal. That influenced tyJC to increase their score.

I therefore recommend accept. The authors shall update their camera ready with all the changes promised in the rebuttal.